# Population Density and Abundance of the Northernmost Population of *Cordulegaster heros* (Anisoptera: Cordulegastridae) in Europe (Czech Republic) with Notes on Its Biogeographical Range

Otakar Holuša [1],* and Kateřina Holušová [2]

1. Department of Environmental Science and Natural Resources, Faculty of Regional Development and International Studies, Mendel University in Brno, Tř. Gen Píky 7, CZ-613 00 Brno, Czech Republic
2. Department of Forest and Wood Product Economics and Policy, Faculty of Forestry and Wood Technology, Mendel University in Brno, Zemědělská 3, CZ-613 00 Brno, Czech Republic
* Correspondence: holusao@email.cz

**Abstract:** *Cordulegaster heros* is a Balkan species with a disjunctive area extending into Central Europe. The population in the Chřiby Mts. in the southeastern Czech Republic is the northernmost population, and this population was intensively studied from 2010 to 2021 to establish basic data on its abundance. In the territory, the geomorphological characteristics of streams, characteristics of sediment in streams, habitat, emergence time, and period of flight were recorded, and population viability was evaluated. Larvae were recorded in 10 small forest streams (altitude of 235–426 m a.s.l.), with an average minimum width of 51.9 cm, an average maximum width of 177.7 cm, an average minimum depth of 6.5 cm, an average maximum depth (in pools) of 21 cm, and an average stream gradient of 1.9 grades. The sediments in each stream exhibited a grain size distribution with an average fraction less than 0.05 mm represented by 6.3%, a fraction of 0.05–0.1 mm represented by 21.1%, a fraction of 0.1–2 mm represented by 52.1%, a fraction of 2–5 mm represented by 12.1%, a fraction of 5–20 mm represented by 8%, and a fraction of 20+ mm represented by 0.3%. The larval abundance was 0.1–6.7 larvae per 1 m$^2$ of suitable sediment. The emergence period was recorded from 28 May to 1 July. The emergence site was categorized as larvae-dominated plant leave (57% of cases), plant stalks (21%), and tree trunks (17%). Exuviae occurred at an average of 154 cm at horizontal distance from the shore and an average vertical height of 77 cm above the ground. The average total distance of larval movement was 205 cm. The flight period in 2021 was recorded from 15 June to 11 August with peak flight activity noted in the third week of June. The northernmost population of *C. heros* was evaluated as viable and stable.

**Keywords:** *Cordulegaster heros*; Cordulegastridae; Odonata; northern border distribution; population abundance; biogeography; Chřiby Hills; Czech Republic

## 1. Introduction

*Cordulegaster heros* was described by Theischinger [1] with type locality in Austria, Lower Austria-Sankt Andrä vor dem Hagnethale in central Europe. The species is found on the Balkan Peninsula (including Greece, North Macedonia, Montenegro, Albania, Croatia, Slovenia, Serbia, Bosnia and Herzegovina, Bulgaria, northeastern Italy, and Romania) and central Europe, including the foothills of the Alps in eastern Austria [2–5], the foothills of the Carpathians Mountains in Slovakia [6,7], the Czech Republic [8], and outside of the Carpathians Mountains in Ukraine [9]. Separate areas of growth have been reported from the hills in the middle of the Pannonian lowland in Hungary [10,11].

Since the species was described approximately 45 years ago, its range has seen great progress in the last few years in terms of knowledge and boundary refinement. In 1988,

practically nothing was known about its distribution, as the species occurrence was known from a few localities in Austria, western Hungary, the western part of the former Yugoslavia [12], and Greece [1]. Later, Boudot [2] reported its distribution over a large area of the Balkans Peninsula, but the range often follows national boundaries and includes "geometric" connections between the northern and southern parts of the range. Later, van Pelt [3] removed some inconsistencies and reported a more or less disjunctive range throughout the Balkans with the northernmost extension in southern Slovakia. The most up-to-date range has been recently reported [13], where the areal distribution throughout the western Balkans is already shown, although with "gaps" of knowledge in Albania and Bosnia and Herzegovina. Unfortunately, there are still insufficient data on the northern and northeastern borders, as evidenced by the findings for the territory of Ukraine [9]. The center of the species range is located in Slovenia, eastern Austria, and northwestern Croatia, where the species is common, with large populations reported ([14,15], Holuša unpubl.). Until the description of *C. heros* in 1979, its occurrence was confused with that of the congeneric species *C. boltonii* and *C. bidentata*. A more precise eastern limit of the distribution of *C. boltonii* in Central Europe was reported by Holuša [16]. Although *C. boltonii* is reported in Romania [17] and Slovakia [18–20], this was possibly due to confusion with *C. heros* and *C. bidentata* [16].

The possible occurrence of *C. heros* in the territory of the Czech Republic was first reported by Bernard (pers. Comm.). Later, Holuša [21] introduced the idea that the species could occur in the southeastern part of the territory of the Czech Republic on the southern slopes of the Carpathian Mountains on the border with the Pannonian lowland. However, extensive surveys of this area since 1998 confirmed the occurrence as late as 2009 (one dead female) [8]. This record has been the absolute northernmost occurrence reported to date. In contrast, Staufer and Holuša [8] "classified" it as an occasional "visit"; nevertheless, they conceded the potential occurrence of a small permanent population. The occurrence of larvae was established in 2011 from one specimen at the locality of the Kudlovický potok stream in the Kudlovická valley near the village of Kudlovice and at the Jankovický potok stream in the village of Jankovice [22].

The aim of this paper is to present a complete overview of all findings of *C. heros* in the Czech Republic during intensive research carried out in 2010–2021 in order to provide a first evaluation of population abundance and to evaluate its stability. Knowledge of the occurrence and ecology of this species is also very important, as this species is listed as requiring protection (Annex II and IV of Council Directive 92/43/EHS).

## 2. Materials and Methods

A detailed survey was performed in the area of the Chřiby Hills and Ždánický les Hills in the southeastern part of the Czech Republic. In both areas, all watercourses were repeatedly surveyed (grid squares of the Central European Mapping Network 6867, 6868, 6967, and 6968 for Ždánický les Hills [23] and 6769, 6869, 6870, 6968, 6969, 6970, and 6770 for Chřiby Hills). After finding larvae in 2011 in the northern area of the Chřiby Hills, some streams were selected for intensive surveys. Several sites were then selected on each stream to survey larvae in detail with individual sites located approximately 500–700 m from each other, depending on the nature of the stream. The following number of sites was then selected on each stream (Table 1). Only sites where *C. heros* was detected are listed in the site descriptions.

**Table 1.** Description of watersheds with localities with *Cordulegaster heros* records in the Czech Republic (only localities with detected occurrence are listed, localities are sorted upstream).

| Number of Watershed | Locality Name (Cadastral Territory) | Code of Locality * | Altitude (m a.s.l.) | Geographical Coordinates (N/E) | | Grid Mapping Square | Larvae Density (per 1 m² of Sediment) |
|---|---|---|---|---|---|---|---|
| I. | Cvrčovický potok stream (Cvrčovice u Zdounek) | Ia. | 235 | 49°13′24.71″ | 17°20′28.89″ | 6770 | 0.7 |
| | | Ib. | 248 | 49°13′17.51″ | 17°20′51.26″ | 6770 | 0.4 |
| | | Ic. | 265 | 49°12′59.78″ | 17°21′08.39″ | 6770 | 0.8 |
| | | Id. | 270 | 49°12′45.78″ | 17°21′15.83″ | 6770 | 0.3 |
| | | Ie. | 294 | 49°12′19.27″ | 17°21′35.88″ | 6770 | 0.7 |
| | | If. | 307 | 49°12′06.05″ | 17°21′23.53″ | 6770 | 0.1 |
| II. | Divocký potok stream (Divoky) | IIa. | 285 | 49°12′11.69″ | 17°19′42.62″ | 6769 | 0.4 |
| | | IIb. | 298 | 49°11′54.36″ | 17°19′50.56″ | 6869 | 0.2 |
| | | IIc. | 312 | 49°11′53.77″ | 17°19′51.13″ | 6869 | 4.0 |
| III. | Roštínský potok stream (Roštín) | IIIa. | 371 | 49°10′46.72″ | 17°18′44.81″ | 6869 | 0.1 |
| IV. | Vašákův potok stream (Roštín) | IVa. | 340 | 49°10′34.67″ | 17°17′23.07″ | 6869 | 0.2 |
| V. | stream of Litava (Zástřizly) | Va. | 361 | 49°08′52.72″ | 17°14′54.34″ | 6869 | 0.3 |
| | | Vb. | 426 | 49°09′05.37″ | 17°16′06.35″ | 6869 | 0,1 |
| VI. | Kudlovický potok stream (Košíky, Lubná u Kroměříže, Kostelany) | VIa. | 255 | 49°10′11.52″ | 17°25′51.31″ | 6870 | 0.7 |
| | | VIb. | 295 | 49°11′01.59″ | 17°24′41.22″ | 6870 | 4.5 |
| | | VIc. | 320 | 49°11′39.34″ | 17°23′54.94″ | 6870 | 0.2 |
| | | VId. | 328 | 49°11′41.59″ | 17°23′26.90″ | 6870 | 0.1 |
| | | VIe. | 320 | 49°11′18.41″ | 17°23′59.90″ | 6870 | 0.3 |
| | Habešský potok stream–right-side tributary of Kudlovický potok stream | VIf. | 346 | 49°11′07.23″ | 17°23′40.57″ | 6870 | 0.3 |
| | | VIg. | 356 | 49°10′59.78″ | 17°23′20.28″ | 6870 | 6.7 |
| VII. | Jankovický potok stream (Jankovice u Uherského Hradiště) | VIIa. | 309 | 49°09′44.99″ | 17°22′36.64″ | 6870 | 0.5 |
| | | VIIb. | 316 | 49°09′49.38″ | 17°22′32.72″ | 6870 | 6.3 |
| | | VIIc. | 334 | 49°09′59.90″ | 17°22′28.30″ | 6870 | 1.3 |
| | | VIId. | 352 | 49°10′07.97″ | 17°22′19.96″ | 6870 | 0.4 |
| | | VIIe. | 355 | 49°10′16.87″ | 17°22′15.19″ | 6870 | 0.1 |
| | | VIIf. | 362 | 49°10′26.88″ | 17°22′04.71″ | 6870 | 0.2 |
| | | VIIg. | 383 | 49°10′35.61″ | 17°21′56.09″ | 6870 | 1.3 |
| | Upper left-side tributary of Jankovický potok stream | VIIh. | 354 | 49°10′21.73″ | 17°22′14.47″ | 6870 | 0.3 |
| | Upper right-side tributary of Jankovický potok stream | VIIi. | 371 | 49°10′21.12″ | 17°21′59.64″ | 6870 | 0.2 |
| VIII. | Bunčovský potok stream (Velehrad) | VIIIa. | 352 | 49°09′53.34″ | 17°20′53.31″ | 6870 | 0.3 |
| | | VIIIb. | 355 | 49°10′07.99″ | 17°20′42.39″ | 6870 | 0.1 |
| | | VIIIc. | 368 | 49°10′05.02″ | 17°20′49.50″ | 6870 | 0.2 |
| | | VIIId. | 403 | 49°10′24.00″ | 17°20′55.09″ | 6870 | 0.1 |
| | | VIIIe. | 286 | 49°09′08.41″ | 17°21′46.88″ | 6870 | 4.0 |
| | | VIIIf. | 389 | 49°10′15.71″ | 17°20′58.41″ | 6870 | 0.1 |
| IX. | stream of Salaška (Salaš u Velehradu) | IXa. | 298 | 49°08′42.51″ | 17°20′15.25″ | 6870 | 0.8 |
| | | IXb. | 319 | 49°09′00.89″ | 17°19′49,33″ | 6869 | 0.1 |
| | | IXc. | 323 | 49°09′06.93″ | 17°19′09.09″ | 6869 | 0.5 |
| | | IXd. | 342 | 49°09′05.95″ | 17°19′19.44″ | 6869 | 1.5 |
| | Upper left-side tributary of stream of Salaška | IXe. | 332 | 49°09′07.25″ | 17°19′53.29″ | 6869 | 5.0 |
| | | IXf. | 344 | 49°09′20.63″ | 17°18′51.26″ | 6869 | 0.1 |
| X. | Buchlovický potok stream (Buchlovice) | Xa. | 310 | 49°05′48.06″ | 17°19′01.18″ | 6969 | 4.0 |

* roman numeral - designation of the watershed of watercourse, alphabetical letter - location on the watercourse

For the watersheds, individual sites (locations) were selected on the main stream and tributaries, where larvae were surveyed over a 50 m transect. In this reach, all suitable sediment deposits where larvae were suspected to occur were examined in detail. Where larvae were found, basic stream characteristics (stream width—minimum/maximum), stream depth (minimum/maximum in pools), and stream gradient were recorded. Phytocenological notes were made on vegetation and tree cover composition, and other dragonfly species were monitored. After the total larvae were found and recorded, they were again released into the stream. At the site where larvae were detected, a sediment sample weighing approximately 700–1000 g was collected, to determine the sediment grain fraction representation. Grain size was determined on the dried sample, which was divided into fines and skeleton fractions. The skeleton was separated into grain size fractions (2–5 mm, 5–20 mm, 20 mm or greater) using sieves and the fine-earth into grain size fractions (less than 0.05 mm, 0.05–0.1 mm, and 0.1–2 mm) using the floating method [24]. For the 50 m transect, the suitable sediment area was estimated using the average length and average width.

Emergence monitoring was conducted on all streams from 2013–2014, and then 2020–2021. A 100 m transect was selected where larvae were detected. Exuviae were collected from the beginning of May to the beginning of July in the riparian parts of the sites studied. The search for larvae occurred approximately 10 m from the shoreline and up to 5 m on tree trunks. The distance from the shoreline (vertical projection of the emergence place) and the height above the ground (vertical distance from the place where the shoreline ends) were measured. The total distance to the emergence place is the sum of the height and the distance from the shore, as travelled by the larvae from the shoreline to the place where the adult emerged. The position of its thorax was used to identify the position of the exuviae, and distances were measured from this point. The site of emergence was evaluated according to where the exuviae was attached, i.e., tree roots, tree trunks, tree sticks, tree leaves, plant stalks, and plant leaves.

In 2021, the flight period of the adults at the Habešský potok stream (locality VIf) stream site was studied in detail. The flight activity of adults was monitored from 2 June to 18 August 2021. The site was monitored in suitable weather (partly cloudy to clear sky, midday temperature, approximately 20 °C or greater) from 8:30 a.m. to 7 p.m. CET from 2 to 15 June, from 15 July to 18 August 2021 (sunrise on these days at this latitude is at 4:55 and 5:20 a.m. EST, respectively), and from 5:30 a.m. to 9:15 p.m. CET from 19 June to 11 July 2021 (sunrise on these days at this latitude ranges from 4:50 to 5:05 a.m. CET). The passing adult was caught on the first pass and marked with a number on the wings. On the next overflight, only the number was detected, and the overflight was noted. The total number of adults within a day and the passage of non-marked adults were recorded. The number of adults observed each day was determined based on the number of marked individuals and the number of passes of unmarked individuals.

The distribution of the species in Europe was processed in a grid map of 12 × 11 km squares to correspond to the KFME (Kartierung der Flora Mitteleuropas) squares for each country. Previously published data [7,25,26] and our own data [Holuša unpbl.] for Slovakia, Hungary [27], and Austria [14] were used to show the occurrence in individual countries. The Pannonian biogeographical boundaries are modified for Slovenia and Austria based on state boundaries but are based on the works of [28,29]. The maps were processed in ESRI 2020 ArcGIS ArcMap 10.8 software.

## 3. Results

### 3.1. Locations of Codulegaster Heros Populations in the Czech Republic

*Cordulegaster heros* was found at 42 localities (Table 1) in 10 forest stream catchments in the northern Chřiby region of southern Moravia in the southeastern part of the Czech Republic (Figure 1) in an area approximately 100 km² in size. These localities are located in five faunistic squares, 6769, 6770, 6869, 6870, and 6969, at altitudes ranging from 235 to 426 m a.s.l. with an average of 328 m a.s.l. (Figure 2). The species was not detected in the adjacent area, i.e., Ždánický les Hills.

The area of occurrence of *C. heros* in Chřiby Hills in the territory of the Czech Republic constitutes a separate area within its disjunctive area. The nearest known occurrence in Austria is 135 km to the southwest ([30], Holuša unpbl.), and the range in the Záhorie lowland and in the Little Carpathians (Malé Karpaty Mts.) in Slovakia [7] (Figure 1) is 50 km to the south. The area in Chřiby Hills is the northernmost occurrence of the species, and the locality Cvrčovický potok stream (Cvrčovice village cadastral territory), with coordinates of 49°13′24.71″ N, 17°20′28.89″ E, is the absolutely northernmost point of occurrence in Europe and the world.

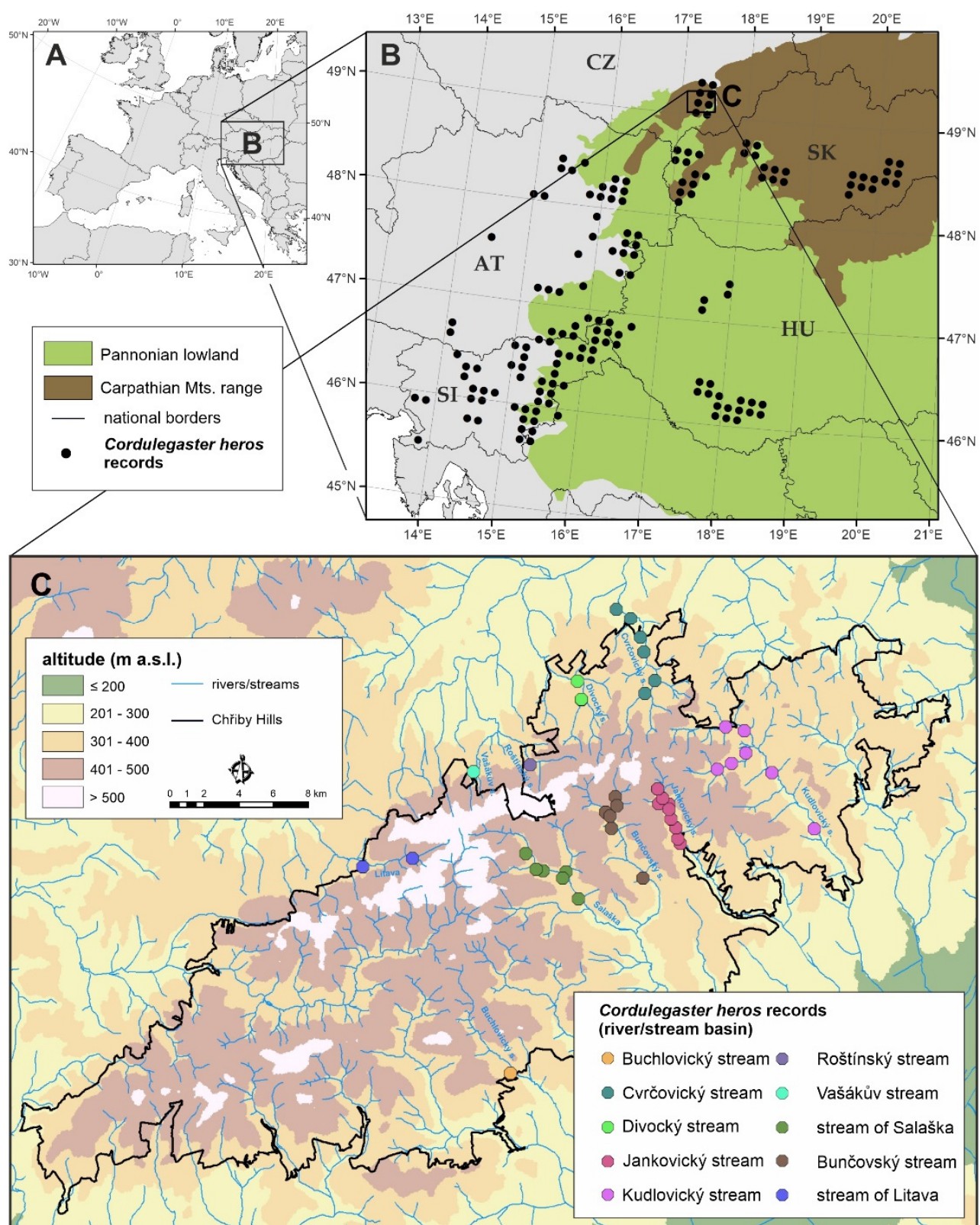

**Figure 1.** Known records of *Cordulegaster heros* in the Czech Republic (state as of 30 June 2022). (**A**) Map of western, south and central Europe (**B**) Map of central Europe with records of *C. heros* (AT—Austria, CZ—Czech Republic, HU—Hungary, SI—Slovenia, SK—Slovakia) (**C**) Region of the Chřiby Hills in the Czech Republic with *C. heros* records in individual watershed.

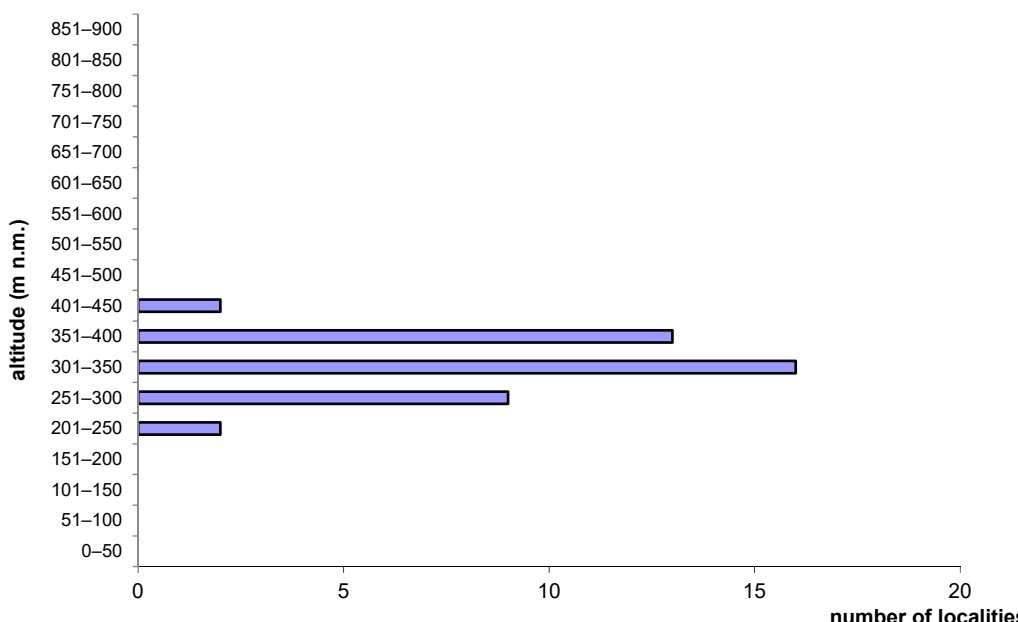

**Figure 2.** Number of localities with *Cordulegaster heros* populations in the Czech Republic based on altitude (all authors (*n* = 42)).

*3.2. Habitat Characteristics*

The species habitat is characterized by forest streams, with an average minimum width of 51.9 cm, an average maximum width of 177.7 cm, an average minimum depth of 6.5 cm, an average maximum depth (in pools) of 21 cm (Figure 3a), and an average stream gradient of 1.9 grades (range 1–3.5 grades). All streams are natural or seminatural, and more or less meandering. In some locations, the stream courses have been straightened due to the construction of forest roads, and the adjacent bank is strengthened in places by stone flatwork. The banks are gradual but also straight where the stream bed meets the adjacent slope. Most of the stream alluvium is covered by a variable vegetation cover (average 46% with values ranging from 5 to 100% based on tree cover).

The vegetation is most frequently composed of the following species listed in order of frequency of occurrence: *Aegopodium podagraria*, *Athyrium filix-femina*, *Carex remota*, *C. sylvatica*, *Dryopteris filix-mas*, *Geranium robertianum*, *Glechoma hederacea*, *Impatiens noli tangere*, *Lamium maculatum*, *Urtica dioica*, *Stachys sylvatica*, *Petasites albus*, *Pulmonaria officinalis*, *Rubus fruticosus*, *R. hirtus*, *Carex brizoides*, *Brachypodium sylvaticum*, *Carex pendula*, *Lycopus europaeus*, *Lysimachia numularia*, *Mercurialis perennis*, *Oxalis acetosella*, *Impatiens parviflora*, and others. All localities are located in the forest complex of Chřiby Hills (Figure 4a,b). The alluvia are covered by forest stands characterized by the following two types of composition: 1. *Alnus glutinosa* and *Fraxinus excelsior* with an admixture of *Fagus sylvatica* and *Acer pseudoplatanus*, and 2. dominated by *Fagus sylvatica* and *Acer pseudoplatanus* with an admixture of *Carpinus betulus* and *Alnus glutinosa*. Individually, *Tilia cordata*, *Quercus petraea*, *Sambucus nigra*, nonnative *Picea excelsa*, and *Larix decidua* may also be represented.

The presence of suitable sediment is necessary for larval survival. Based on grain size characteristics (Figure 3b), 6.3% of samples exhibit an average fraction of less than 0.05 mm, 21.1% with a fraction of 0.05–0.1 mm, 52.1% with a fraction of 0.1–2 mm, 12.1% with a fraction of 2–5 mm, 8% with a fraction of 5–20 mm, and 0.3% with a fraction of 20+ mm. Thus, the average ratio of fine (fraction less than 2 mm grain size) to skeleton (fraction greater than 2 mm) sediment was 79.6:20.4%. This finding shows that the sediment in these habitats is sandy and dominated by medium sand with an admixture of fine sand and a mixture of gravel. Stones (from 20 mm) and clay are very poorly represented.

In habitats with *C. heros*, seven additional dragonfly species were found. The most frequent three species included *Calopteryx virgo*, *Cordulegaster bidentata* (larvae of both

species were found), and *Aeshna cyanea*. Only single adult specimens of other species, including *Ophigomphus cecilia*, *Somatochlora metallica*, *Platycnemis pennipes,* and *Lestes viridis*, were recorded. The larvae of *C. bidentata* were recorded syntopically at several sites at the small side tributaries of the main streams, where *C. bidentata* occurs. The larvae enter the main stream by flushing at higher water levels in the watercourse.

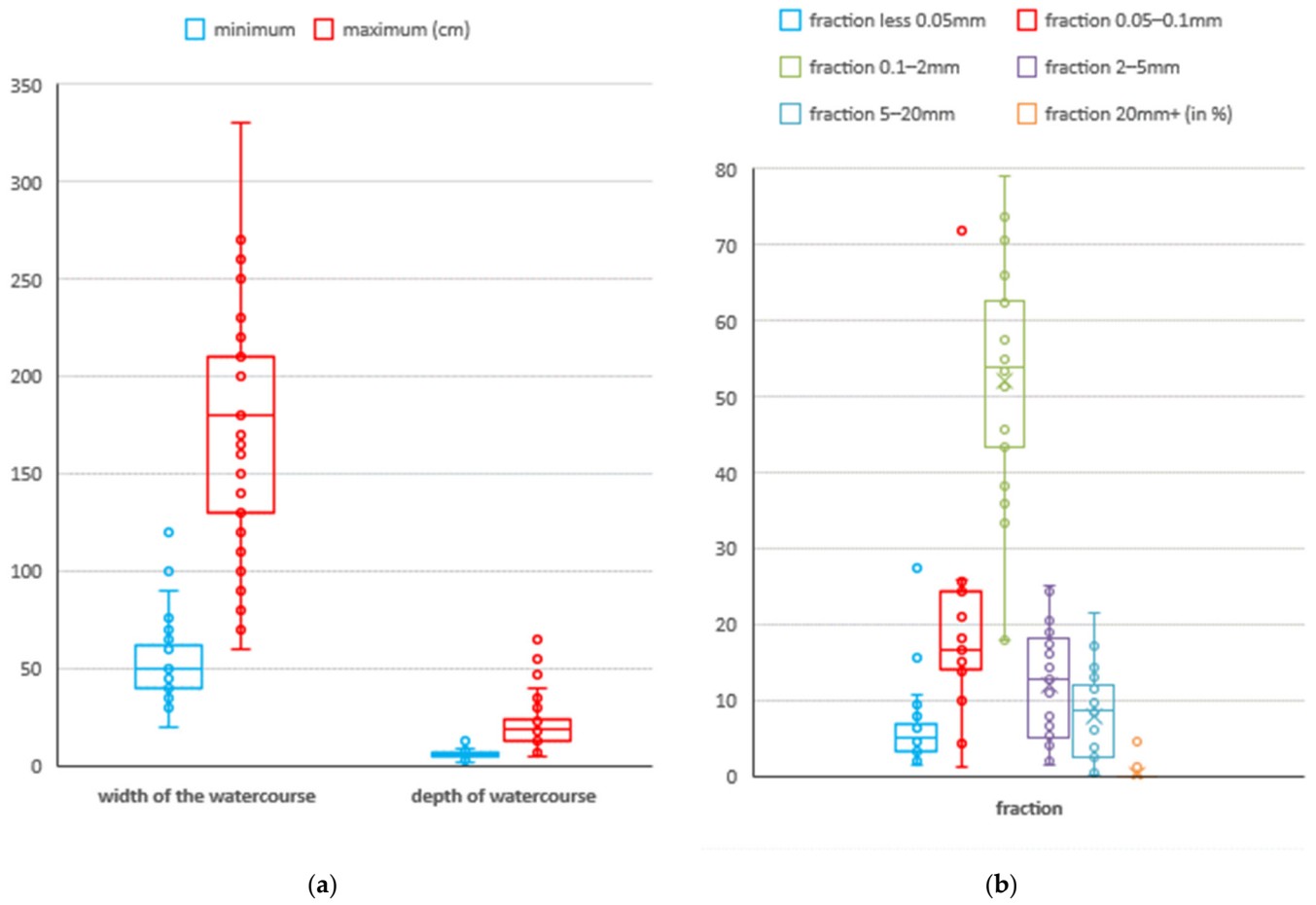

(**a**)　　　　　　　　　　　　　　　　　　　　　　　　　　　　　　　(**b**)

**Figure 3.** Characteristics of habitat at localities with *Cordulegaster heros* larvae (*n* = 42): (**a**) width and depth of watercourses, (**b**) grain composition of the sediment.

### 3.3. Population Abundance

The larvae of *C. heros* occurred in areas with suitable sediment, which varied from single point locations (e.g., locality Xa) to beds that were approximately 100% composed of alluvial sediment (e.g., locality VIIIb). The average larval abundance (all instars combined) was 2.8 larvae/50 m section. With respect to sediment area, the density reached 0.1–6.7 larvae per 1 m$^2$ of sediment (Table 1). The maximum number of larvae found per 50 m section of watercourse was 10 with a maximum of 4 larvae per site (15 × 15 cm). It is clear from the distribution of sites that most larvae were found in the upper parts of the streams (e.g., Kudlovický potok stream); thus, larvae found in the lower parts of the streams are due to larval flushing during high flow conditions. In these lower parts, the occurrence of adults (both males and females) was recorded, but no ovipositions were observed. Females most often prefer streams with specific characteristics (Figure 3a,b).

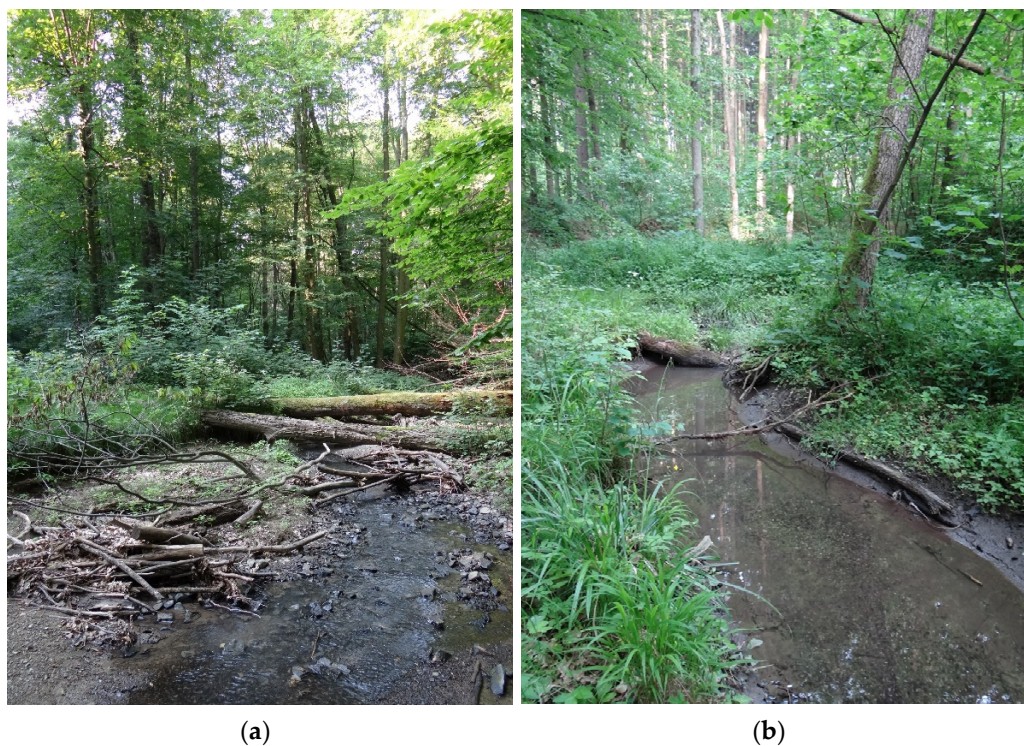

(**a**)  (**b**)

**Figure 4.** Habitat of *Cordulegaster heros* in the Czech Republic: (**a**) locality of Habeššký potok stream (Kostelany cadastral territory, southern Czech Republic) (locality VIe) (18 July 2013, photo Otakar Holuša), (**b**) locality of Cvrčovický potok stream (Cvrčovice cadastral territory, southern Czech Republic) (locality Id) (4 June 2016, photo Otakar Holuša).

### 3.4. Time and Place of Emergence

The emergence period occurred from 28 May (the earliest date of detection of exuviae in 2014) to 1 July (the latest date of detection of exuviae in 2013). A total of 186 exuviae were found (several exuviae were in varying degrees of damage, such as the abdomen or head were missing). The site of emergence was evaluated with respect to location, i.e., the place where the adult emerged: (a) tree roots (exposed roots of living trees in the vertical banks of the streams), (b) tree trunks, (c) tree sticks, (d) tree leaves, (e) plant stalks, and (f) plant leaves (Figure 5). The presence of these emergence places is based on the characteristics of the habitat that the species inhabits, namely, watercourses shaded with rich vegetation and tree layers. Plant leaves were the dominant site of emergence (57%, Figure 5) followed by plant stalks (21%) and tree trunks (17%), tree leaves (15%), tree sticks (5%), and tree roots (4%) (Figure 6a,b). Even the roots of old trees, which were uncovered and formed "overhangs" from the soil due to water erosion, were selected by larvae for emergence. It is obvious that the larva at the end of its "journey" climbed as if on the ceiling and continued horizontally on the overhanging root.

Emergence sites were located at different distances from the shoreline and at different heights. Specifically, exuviae were noted from 0 to 790 cm ($\ddot{x}$ = 154 cm) at a horizontal distance from the shore and from 10 to 310 cm ($\ddot{x}$ = 77 cm) at a vertical height above the ground. The total distance of larval movement ranged from 10 to 1080 cm ($\ddot{x}$ = 205 cm) (Figures 7 and 8).

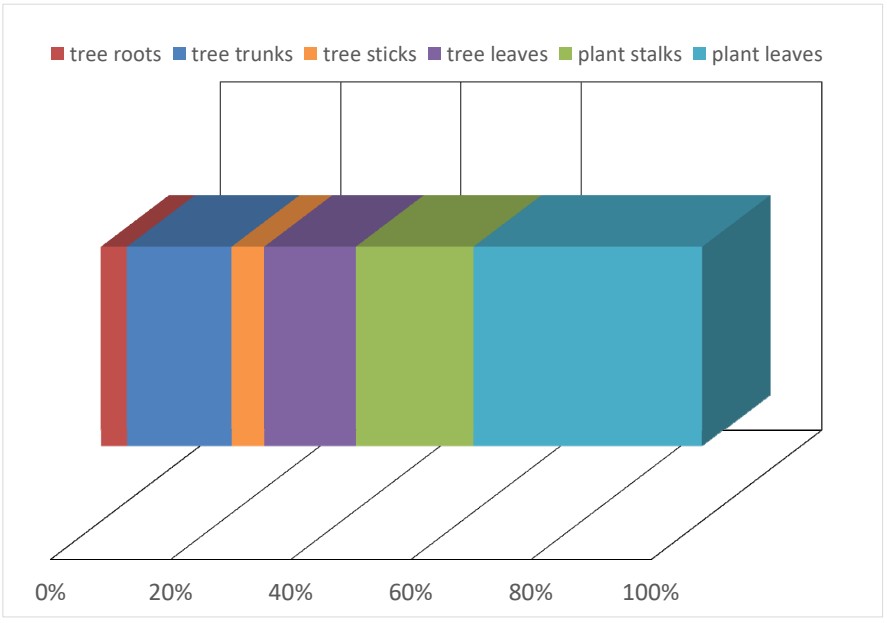

**Figure 5.** Proportion of different types of emergence sites of *Cordulegaster heros*.

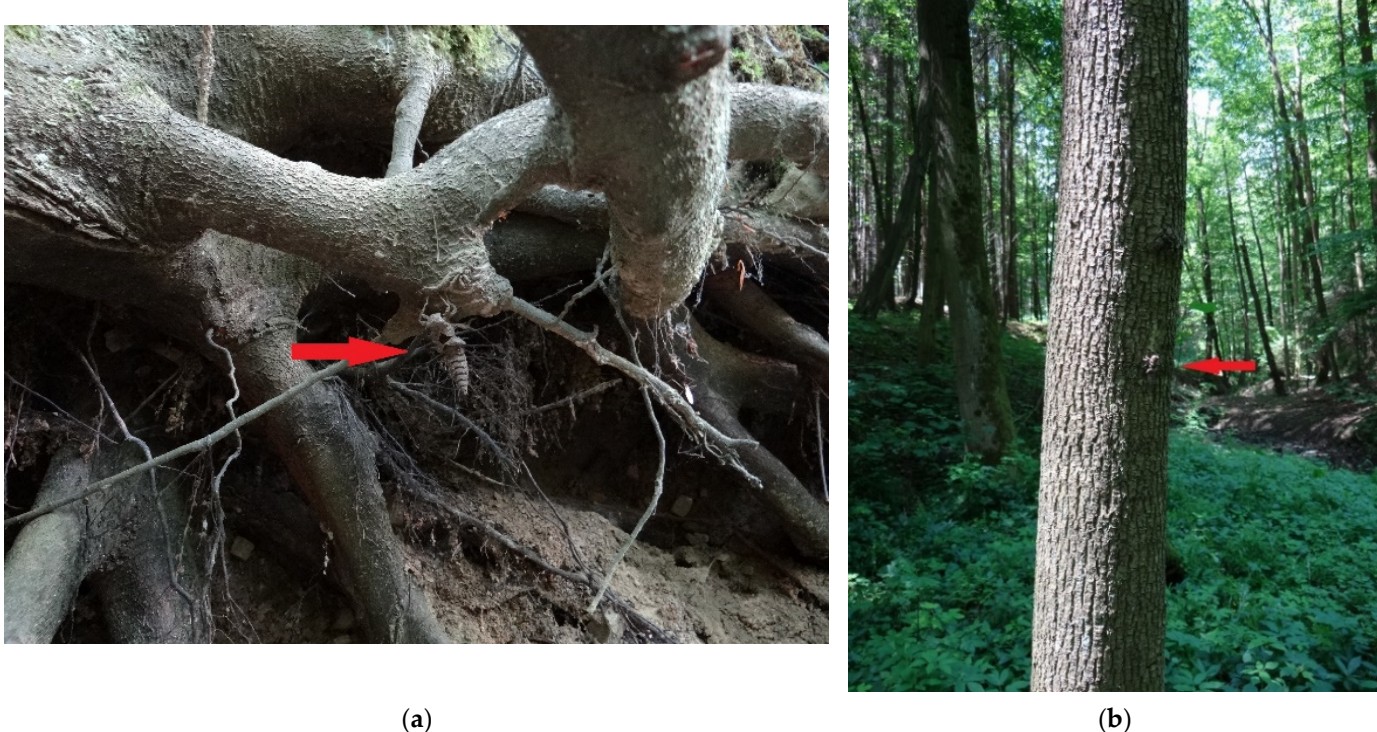

(**a**)                    (**b**)

**Figure 6.** Position of exuviae of *Cordulegaster heros*: (**a**) on tree roots at steep bank, locality Cvrčovický potok stream (Cvrčovice cadastral territory, southern Czech Republic) (locality Ie), (4 June 2016, photo by Otakar Holuša); (**b**) on trunk of *Alnus glutinosa*, locality Cvrčovický potok stream (Cvrčovice cadastral territory, southern Czech Republic) (locality Ie), (8 July 2016, photo by Otakar Holuša).

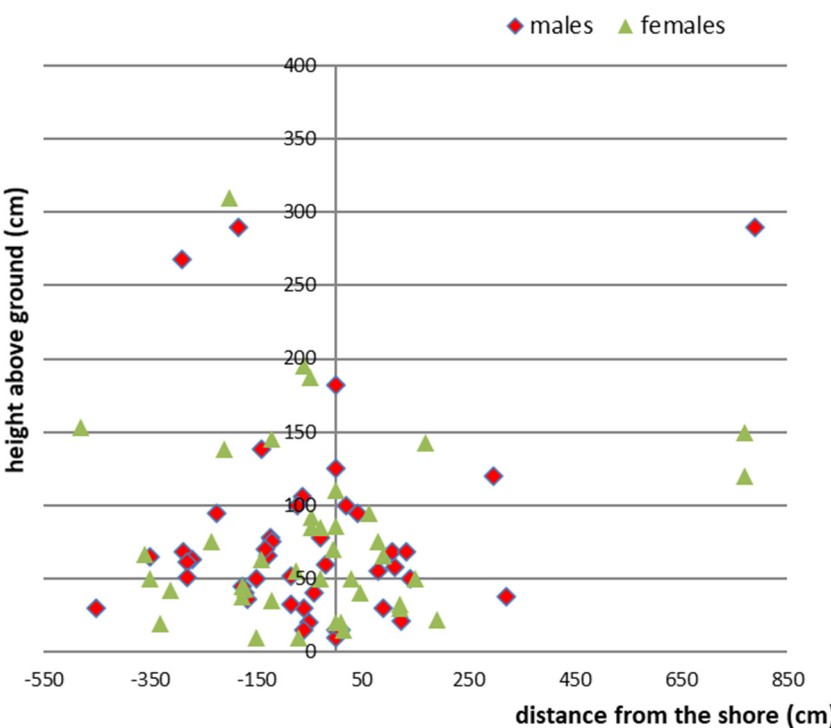

**Figure 7.** Position of exuviae of *Cordulegaster heros* in riparian parts of the habitat (x-axes: (−) left bank, (+) right bank).

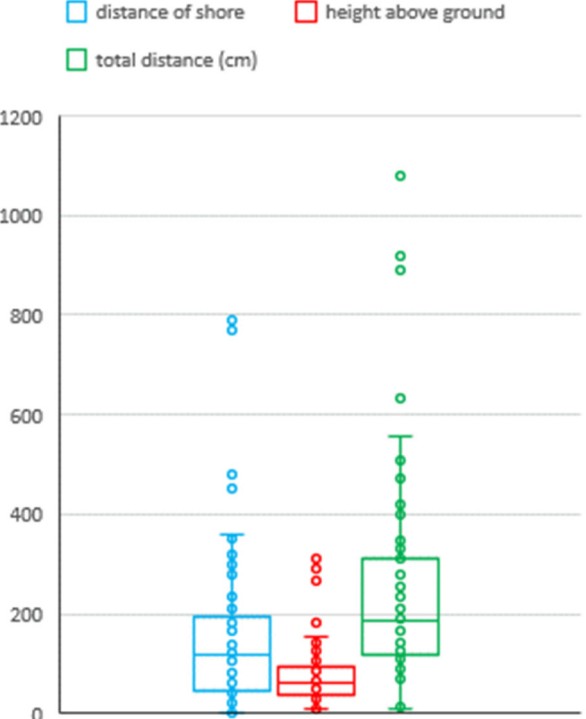

**Figure 8.** Variability of position of exuviae of *Cordulegaster heros* (distance from shore, height above ground, and total distance; all reported in cm).

### 3.5. Flight Time of Adults

The flight period of *C. heros* was identified from 15 June 2021 (the earliest date of detection of adults outside the year of detailed flight period monitoring) to 11 August 2021 (the latest date of detection of flying adults was 13 August 2013; one male observed)

(Figure 9). Flying activity increased sharply in the second half of June, with a peak in the second half of June. Specifically, on 23 June 2021, 103 individuals were recorded during the day plus another 150 flights unmarked adults. Thus, the potential number of adults in one day at one site was approximately 200. In the second half of June, flight activity began to decline gradually, with ca. 40 individuals (marked) and an additional 30 overflights observed in early July. In the second half of July, a further decline was noted with only 3 to 10 individuals (marked) recorded per day with 10 additional overflights. In the first ten days of August, only a few individuals were observed (four to six individuals in total). The flight period was completed at the end of the first third of August. No additional flying individuals were recorded after 13 August. Based on the comparison of the emergence and flight periods, it is obvious that adults spend approximately 2 weeks out of the habitat after emergence before returning and starting their activity by flying in their "home" habitat.

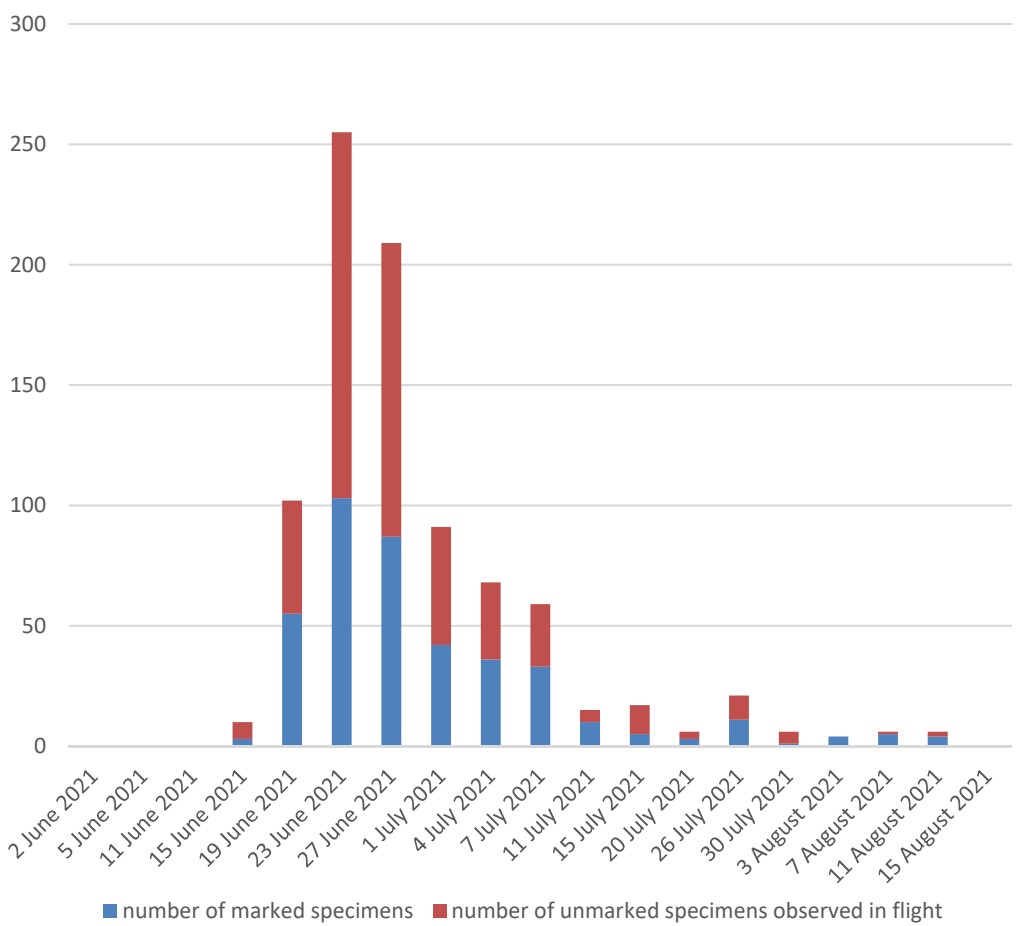

**Figure 9.** Flight time of *Cordulegaster heros* in the Habešský potok stream (Kostelany cadastral territory, southern Czech Republic) (locality VIe) in 2021.

## 4. Discussion

*Cordulegaster heros* is a European endemic species with a habitat that extends throughout the Balkan Peninsula. In the literature, the species is placed in the "east Mediterranean species" group [7], which is more associated with the zoogeographical division of the *boltonii* group of *Cordulegaster* species. As noted in [9], the designation "east Mediterranean" is rather broad given that the eastern part of the Mediterranean is occupied by *Cordulegaster* species related to *C. heros,* i.e., *C. picta*, and further east by *C. vanbrinkae* [31] and the currently least known *C. kalkmani* [32].

The species' area is unlikely to experience the same "evolutionary" changes that have occurred over the past 45 years, i.e., the period from the discovery of *C. heros* to the present day. If we compare the state of knowledge in 1988 [12] and today [13], we note great

progress in the knowledge of its distribution. The species' area, thus, mostly includes the countries of the Balkan Peninsula with its boundaries lying in the northeastern area of Italy, eastern Austria, southeastern Czech Republic, southern Slovakia, western Ukraine, and eastern Romania. On this boundary, the occurrence is not continuous, comprising several separate areas. The northernmost area is the Chřiby Hills area in the Czech Republic (Figure 1), with the closest population occurring in Austria at a distance of 160 km and the easternmost population occurring in Slovakia at a distance of 70 km. Thus, the absolute northernmost point of occurrence is the locality of the Cvrčovický potok stream in Cvrčovice village at GPS 49°13′24.71″ N and 17°20′28.89″ E. The species is not thought to live outside of the surveyed area in the Chřiby Hills, as the landscape further afield is an agricultural landscape without forest belts along the streams.

Taking into account the distribution of the species and its center of the area, which is due to its numerous occurrences in Slovenia ([15], Bedjanič pers. comm., Holuša unpubl.), northern Croatia (Holuša unpubl.), and the Illyrian area, *C. heros* can be included in the Illyrian faunal element. Bernard and Daraż [9] refer to this species as part of the Ponto-Mediterranean faunal element, which does not fully correspond to its distribution or its postglacial distribution. From the mentioned Illyrian refugium, it is assumed that the main species of central European forests, such as *Quercus petraea*, *Carpinus betulus*, *Fagus sylvatica*, and *Abies alba*, especially the spreading of vegetation belts sensu Schmid [33,34], used this refugium to survive the ice age. As described by Bernard and Daraż [9], *C. heros* used three migratory directions to inhabit the central European area in the postglacial period: western (going north along the Austrian–Hungarian border from the refugium), central (going northeast along the eastern foothills of the Carpathian Mountains in western Romania), and eastern (going north along the eastern foothills of the Carpathian Mountains in eastern Romania towards the Ukraine) directions. The territory of the Czech Republic was, therefore, colonized by the western migration direction, which went from the refugium directly northwards along the eastern foothills of the Alps and the western edge of the Pannonian lowland. This migration stream continued to southern Slovakia, although the *C. heros* area in the space, including eastern Austria, the Czech Republic, and western Slovakia, is disjunctive.

The occurrence of *C. heros* in the Czech Republic has long been presumed by Holuša [16,21], and it has been expected in regions with suitable habitats, i.e., in the belt of regions on the forested southern slopes of the Carpathians Mountains, especially regions without interruption of large lowlands. Although research has been carried out since 1998, the first occurrence of the species in this country was recorded as late in 2009. This first record of *C. heros* adult in 2009 was considered to be a vagrant specimen [8], but the authors did not exclude the occurrence of a permanent population in the Czech Republic. The fact that the population of *C. heros* escaped attention until the beginning of the 21st century can be attributed to two main facts: first, very limited information about the occurrence of central European species of the genus *Cordulegaster* was available, i.e., *C. boltonii* and *C. bidentata,* in the Czech Republic at the end of the century and these species were, therefore, evaluated as "rare" [35–38]. Forest complexes, especially forest springs, were not the focus of attention for entomologists and odonatological research. It was only discovered in the last 20 years that the species are common in the Czech Republic and occupy large areas there (Holuša unpubl.). In addition, there is confusion regarding species determination. The second is the so-called the "paradox of Chřiby Hills". Specifically, the area of the Ždánický Forest Hills and Chřiby Hills has not been and is not currently the focus of attention of entomologists. This area includes small hills with complexes of "monotonous" forests, and the most interesting species are mostly found at in the southern foothills of this area, where sub-Mediterranean species occur, e.g., *Tibicina haematodes* near Archlebov [39]. Moreover, the highest positions of this area do not reach higher altitudes to host montane Carpathian species, so the interest of entomologists has been concentrated in the neighboring areas. It was not until the end of the 20th century that the first odonatological surveys were

performed here [40]; however, *C. heros* was not found here, or, it is possible, it was confused with *C. bidentata*.

The area of Central Europe and the territory of the Czech Republic represents the locations where three species of *Cordulegaster*—*C. boltonii* and *C. heros* of the *boltonii* group and *C. bidentata* as a representative of the *bidentata* group—meet. The occurrence of *C. boltonii* extends its eastern limit of distribution into the Waldviertel in Lower Austria in Austria [14], and localities with syntopic occurrences of *C. boltonii* and *C. heros* are known in this area around Melk [30]. The eastern boundary of the range of *C. boltonii* runs northeast into the Czech Republic in the area of the Českomoravská vrchovina Hills, and then continues along the edge of the Nízký Jeseník Hills (Holuša unpubl.) and continues into Poland, where it follows the foothills of the Carpathian Mountains and continues eastwards towards Ukraine [9]. All the revised data on *C. boltonii* obtained from the territory of Slovakia [20] were assessed as erroneous, as these samples were actually *C. bidentata* [16]. Similarly, *C. boltonii* was reported in the territory of Hungary [41] and Romania [17,42]. In Hungary, it was confused with *C. heros* [43], which is similar to that noted in Romania. In these territories, there is no overlap of the ranges of *C. boltonii* and *C. heros*, and their area boundaries are at least 70 km apart.

The locations of *C. heros* in the Czech Republic have similar characteristics to those in neighboring countries. The altitude fully corresponds to the data from Slovakia, where the species occurs between 160 and 516 m a.s.l., with its main occurrence noted in the range of 201–300 m a.s.l. (43% of all localities) [7]. In Austria, *C. heros* inhabits localities from 180 m a.s.l. to 720 m a.s.l. [14]. In Slovenia, *C. heros* inhabits localities from 0 to 800 m a.s.l. with a range of 200–300 m a.s.l. [15].

The geomorphological characteristics of the streams are comparable to those reported in other areas. In Slovakia, biotopes are reported [7,44,45] as shaded streams and streamlets in forest areas in hills with clean water, with widths from 20 to 420 cm, water depths from 2 to 18 cm, and sandy-gravel sediments. Similarly, Bernard and Daraż [9] from Ukraine describe them as streams with widths of 0.8–1.2 m, sandy-gravel sediment (sand 79.6%, gravel 15.6%), a small admixture of silt (4.8%), and a dominant soil fraction of very fine sand (11.8%) and fine sand (25.6%). From sites in Austria [46], the most favorable microbiotope is characterized as streams with fine, medium, and coarse sand substrate, whereas the less favorable microbiotope is characterized as streams with fine, medium, and coarse gravel substrate. Detailed characteristics of the habitat from the southern part of the species area are not currently available.

With regard to larval density, data are available from Lang [46] and Boda et al. [47]; however, these data are reported in larval density per 10 m of stream length with Hungarian data, indicating larval density per $m^2$. It is not clear what area of suitable sediment was available in these habitats, so comparisons are not possible. The highest local density of 25 larvae per 0.25 $m^2$ of sediment (majority of last larval instar-14) when a group of larvae was collected in a suitable pool was reported from the Little Carpathians (Malé Karpaty Mts.) by Holuša and Kúdela [7].

The observed emergence period is similar to data obtained from more southern areas (Austria) with emergence occurring from the beginning of June to the end of June [14] and southern Hungary from the beginning of the last half of May until 31 July [48]. At sites in Hungary, the locations where exuviae were found are clearly dominated by trees followed by shrubs and plants [46], which is due to the nature of the vegetation of the streambeds. Completely identical results are available for the location of exuviae in the vertical and horizontal directions from the shoreline, including the total distance reported by Boda et al. [48], even with a maximum total distance of 1030 cm, which is the distance the larva had to travel from the shoreline to the emergence site.

The flight activity of *C. heros* is significantly shorter than that of other central European *Cordulegaster* species. However, the data available from Austria [30] and Slovakia [49] are similar to those reported for the northernmost population; thus, the first adults appear in mid-June and fly until the end of July. However, in our case, the adults did not appear

until mid-August. The same peak of flight activity was found by Balász [49] in southern Slovakia. Specifically, in the third week of June, up to 100 individuals per day appeared at the locality.

## 5. Conclusions

Although the northernmost distribution of *C. heros* has an area of approximately 100 km$^2$, it includes 10 individual forest streams and brook watersheds. The populations in the individual watersheds are numerous and do not represent individual records for larvae or adults. However, their numbers are similar to those of populations found further south in Austria and Hungary. In most streams, there is a sufficient abundance of suitable sediment for larval development. The sites are located in forest complexes with a predominance of *Fagus sylvatica*, *Carpinus betulus*, *Quercus petraea*, *Acer pseudoplatanus*, and *Alnus glutinosa*, and thus are forests with a high degree of naturalness. Thus, minimal potential threats are encountered. Existing forests are managed in a manner that does not affect the population status; therefore, a minimal threat is observed. With forest management, only minimal impacts can be expected from timber harvesting and transport across the stream bed. The potential for the construction of water works (small dams or ponds) or the construction or repair of forest roads along streambeds represent significant impacts. The habitat condition is, therefore, excellent, providing suitable conditions for the future occurrence of the population. However, it is necessary to monitor the response of the population to any major interventions in the catchment areas of individual streams and, where appropriate, to monitor the trends in population abundance.

**Author Contributions:** Conceptualization, O.H. and K.H.; methodology, O.H. and K.H.; software, O.H.; validation, O.H.; formal analysis, O.H. and K.H.; investigation, O.H. and K.H.; resources, O.H. and K.H.; data curation, O.H. and K.H.; writing—original draft preparation, O.H. and K.H.; writing—review and editing, O.H. and K.H.; visualization, O.H. and K.H.; supervision, O.H.; project administration, O.H. and K.H.; funding acquisition, O.H. and K.H. All authors have read and agreed to the published version of the manuscript.

**Funding:** This research was funded by Agency for Nature and Landscape Conservation of the Czech Republic of in frame of monitoring NATURA 2000 species.

**Institutional Review Board Statement:** Not applicable.

**Data Availability Statement:** Not applicable.

**Acknowledgments:** We thank to Jiří Trombík (Czech Agricultural University in Prague) for helping with creating map images.

**Conflicts of Interest:** The authors declare no conflict of interest. The funders had no role in the design of the study; in the collection, analyses, or interpretation of data; in the writing of the manuscript; or in the decision to publish the results.

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
