# Peer review of "Population Density and Abundance of the Northernmost Population of Cordulegaster heros (Anisoptera: Cordulegastridae) in Europe (Czech Republic) with Notes on Its Biogeographical Range"

_diversity, doi:10.3390/d14100854_

Round 1

Reviewer 1 Report

The evaluated paper presents very valuable data that were well collected and interestingly commented. I recommend publishing the paper, however, formal, linguistic and content-related defects should be removed first.

I support the authors' arguments that knowledge about the biology and habitat preferences of this particular species and in this particular area is very important. The thoroughness of their examination and the expertise with which the examined elements were selected deserve recognition. This data is unique. It is worth noting that in some Central European countries Cordulegaster heros was included in the high threat categories on the Red Lists, e.g. in Austria it is the EN category. This raises the importance of the presented data.

I have relatively few objections and questions regarding the merits.

The authors discuss in detail the habitat features of the sites with Cordulegaster heros. It is a pity that the stretches of the same watercourses where this species did not occur were not analyzed in a similar way. In that case, one could indicate the features of the habitat that are absolutely unfavourable to the species.

The parameters of the sites of emergence of females and males are compared. It is worth checking whether the differences between them are statistically significant. The standard Mann-Whitney U test is sufficient.

Data sources for mapping the species distribution at the northern edge of the range should be revised and possibly supplemented. I know, of course, that the first author is a principal investigator of the distribution and ecology of Cordulegastridae in the Czech Republic and Slovakia – but after 2010 there were published at least two publications with data from Slovakia in which he did not participate (details in the manuscript). There is also a website vazky.sk, where sometimes unpublished data appear. Raab & Chovanec (2006) as the only basis for creating a map for Austria is also too old. There should be quite a lot of newer data from the last 14 years. The data source from Hungary is newer (from 2017), but I would also do a literature research here. Unfortunately, the creation of such maps and analyzes is tedious, because in the absence of fresh monographic studies, all newer literature has to be carefully searched.

In the "Discussion", information about the habitats of Cordulegaster heros in Central European species are compared. Are there similar data from the southern part of the area of occurrence?

The quality of the English language is a serious drawback of the assessed paper. For me the manuscript is fully understandable, which may be due to the fact that I am also a Slav. However, the number of grammatical, stylistic and spelling errors is unacceptable. The text requires careful correction by a specialist, preferably by a native speaker.

Below I discuss a few minor matters.

Since the text is in English, I would use English geographic names. German names may be entered in brackets if necessary.

Generic and species names should be consistently italicized, and there should be a dot everywhere in decimal fractions (there are commas in Table 1).

The plural of "imago" is "imagines", not "imagoes". Since we use Latin terminology, let us also use Latin grammar. If we want to write in English, we can use the word "adult" / "adults". I know, of course, that "imagoes" appears more and more regularly, especially in publications from the US, but it is worth sticking to the right rules.

The citation method should be carefully standardized in "References". For example, some names of journals are abbreviated, some are given full. If you already use abbreviations, let them not always be appropriate (cf. Holuša 2022). Some monograph titles from which chapters are cited are in italics, and some are not.

Some comments and corrections can be found in the attached manuscript.

Author Response

All comments were accepted and added to the text. The English text has been edited by native speake

Reviewer 2 Report

suggestions given in Track Changes are just suggestions

Author Response

All comments and corrections were accepted and corrections were made to the text. English language has been revised by native speakers.